# Exploration of drug resistance mechanisms in triple negative breast cancer cells using a microfluidic device and patient tissues

Wanyoung Lim[1†], Inwoo Hwang[2†], Jiande Zhang[3], Zhenzhong Chen[3], Jeonghun Han[3], Jaehyung Jeon[3], Bon-Kyoung Koo[4], Sangmin Kim[5], Jeong Eon Lee[6], Youngkwan Kim[2], Kenneth J Pienta[7], Sarah R Amend[7], Robert H Austin[8], Jee-Yin Ahn[2,9,10]*, Sungsu Park[1,3,11]*

[1]Department of Biomedical Engineering, Sungkyunkwan University, Suwon, Republic of Korea; [2]Department of Molecular Cell Biology, Sungkyunkwan University School of Medicine, Suwon, Republic of Korea; [3]School of Mechanical Engineering, Sungkyunkwan University, Suwon, Republic of Korea; [4]Institute of Molecular Biotechnology of the Austrian Academy of Sciences (IMBA), Vienna Biocenter (VBC), Vienna, Austria; [5]Department of Breast Cancer Center, Samsung Medical Center, Sungkyunkwan University School of Medicine, Seoul, Republic of Korea; [6]Division of Breast Surgery, Department of Surgery, Samsung Medical Center, Sungkyunkwan University School of Medicine, Seoul, Republic of Korea; [7]The Cancer Ecology Center at the James Buchanan Brady Urological Institute, Johns Hopkins School of Medicine, Baltimore, United States; [8]Department of Physics, Princeton University, Princeton, United States; [9]Single Cell Network Research Center, Sungkyunkwan University School of Medicine, Suwon, Republic of Korea; [10]Samsung Biomedical Research Institute, Samsung Medical Center, Sungkyunkwan University School of Medicine, Seoul, Republic of Korea; [11]Institute of Quantum Biophysics (IQB), Sungkyunkwan University, Suwon, Republic of Korea

*For correspondence:
jeeahn@skku.edu (JYA);
nanopark@skku.edu (SP)

[†]These authors contributed equally to this work

Competing interest: The authors declare that no competing interests exist.

**Abstract** Chemoresistance is a major cause of treatment failure in many cancers. However, the life cycle of cancer cells as they respond to and survive environmental and therapeutic stress is understudied. In this study, we utilized a microfluidic device to induce the development of doxorubicin-resistant (DOXR) cells from triple negative breast cancer (TNBC) cells within 11 days by generating gradients of DOX and medium. In vivo chemoresistant xenograft models, an unbiased genome-wide transcriptome analysis, and a patient data/tissue analysis all showed that chemoresistance arose from failed epigenetic control of the nuclear protein-1 (NUPR1)/histone deacetylase 11 (HDAC11) axis, and high *NUPR1* expression correlated with poor clinical outcomes. These results suggest that the chip can rapidly induce resistant cells that increase tumor heterogeneity and chemoresistance, highlighting the need for further studies on the epigenetic control of the NUPR1/HDAC11 axis in TNBC.

## eLife assessment

This study based on the use of Cancer Drug Resistance Accelerator (CDRA) chip is **valuable** as a platform technology to assess chemoresistance mechanisms. The strength is **convincing** from the technological point of view. However, the use of a single cell line model is a limitation. However we

acknowledge the authors' plan to further validate their current findings across multiple TNBC cell lines.

## Introduction

A leading cause of cancer-related death is drug resistance (*Jazaeri et al., 2005*), which is increased by tumor heterogeneity (*Heppner and Miller, 1983*). Microfluidic chips are highly useful for studying drug resistance because they can manipulate and control fluids and particles at the micron level (*Yeo et al., 2011*). Recently, a microfluidic platform consisting of an array of connected microchambers with concentration gradients has been developed to induce drug resistance in various types of cancers, such as triple negative breast cancer (TNBC) (*Han et al., 2019*; *Wu et al., 2013*), glioblastoma multiforme (GBM; *Han et al., 2016*), and prostate cancer (*Lin et al., 2020*). In previous studies, we identified the molecular mechanisms involved in doxorubicin (DOX) resistance in GBM and TNBC by analyzing mutation and expression data from chemoresistant cancer cells (*Han et al., 2016*; *Han et al., 2019*). Recently, Lin et al. used a microfluidic chip that generates a docetaxel gradient to induce resistant cells from PC-3 prostate cancer cells (*Lin et al., 2020*). However, the underlying mechanisms by which cells acquire chemoresistance and whether cells obtained from a chip resemble those found in patient tissues remain unknown.

In this study, we utilized the Cancer Drug Resistance Accelerator (CDRA) chip (*Han et al., 2016*) to generate gradients of DOX and medium to induce DOX-resistant (DOXR) cells from MDA-MB-231 TNBC cells within 11 days. Interestingly, a subpopulation of very large cells, referred to as L-DOXR cells, emerged within the DOXR cell population in the CDRA chip on day 11. These L-DOXR cells were

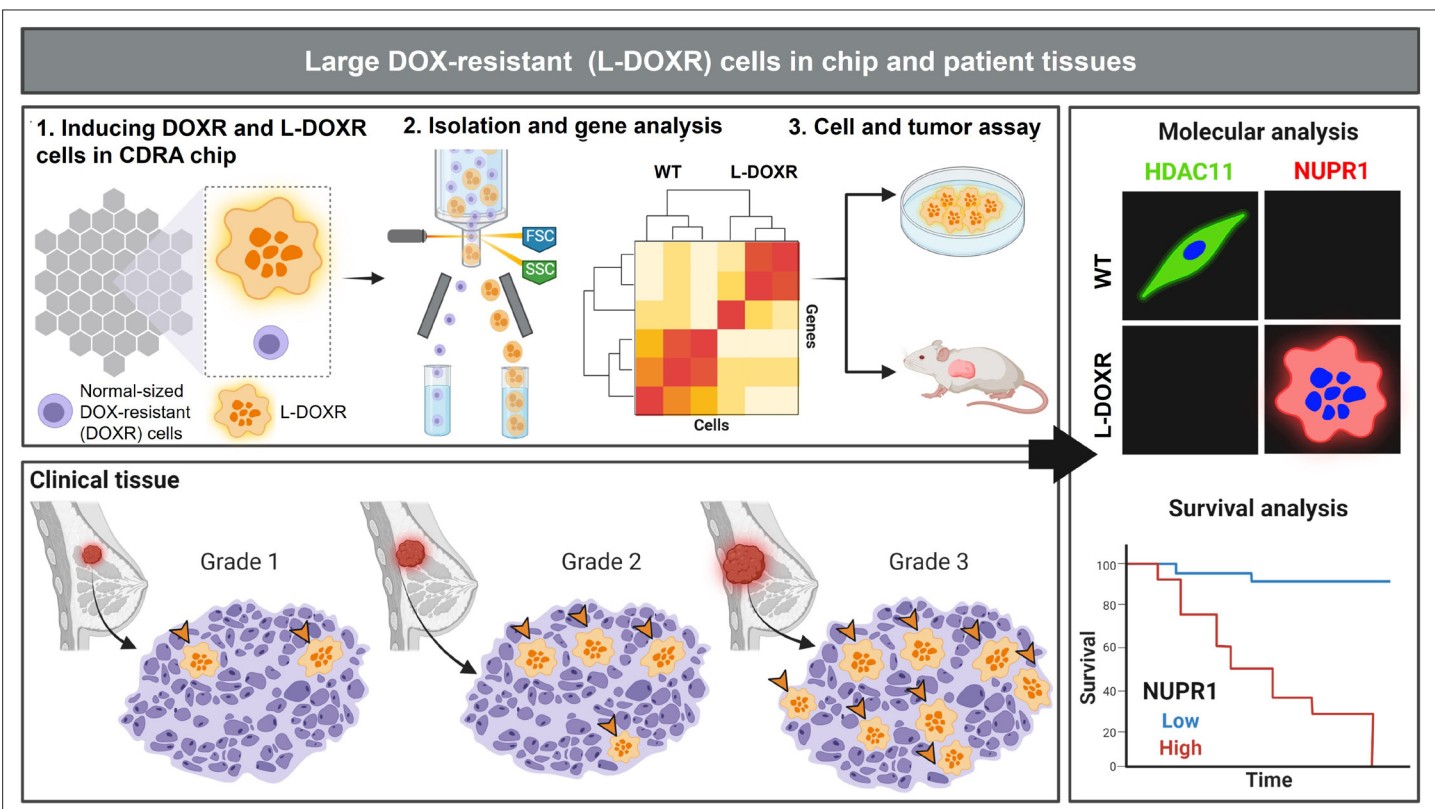

**Figure 1.** Experimental design and analysis workflow. Triple negative breast cancer (TNBC) cells were subjected to doxorubicin (DOX) and nutrient gradients to induce DOX-resistant TNBC cells in a Cancer Drug Resistance Accelerator (CDRA) chip (*Han et al., 2016*). Large DOX-resistant (L-DOXR) cells were sorted by fluorescence-activated cell sorting (FACS) and their transcriptome was analyzed by RNA sequencing (RNA-seq). The oncogenic properties of L-DOXR cells were evaluated in vitro and in vivo to better understand their effect on cancer progression. Additionally, the proportion of L-DOXR cells in TNBC patient tissues was positively associated with TNBC tumor grade. The roles of histone deacetylase 11 (HDAC11) and nuclear protein 1 (NUPR1) in DOX-resistance were investigated through molecular analysis and survival analysis of patients with high/low NUPR1 expression.

isolated using fluorescence-activated cell sorting (FACS) and maintained their survival off the chip. To better understand the role of L-DOXR cells in chemoresistance in TNBC, we conducted in vivo chemo-resistant xenograft models, an unbiased genome-wide transcriptome analysis, and a patient data/tissue analysis. Our results demonstrate that the chemoresistance of L-DOXR cells is attributed to failed epigenetic control of nuclear protein-1 (NUPR1)/histone deacetylase 11 (HDAC11) axis, which can be alleviated through *NUPR1* inhibition (**Figure 1**).

NUPR1, which is also known as Com-1 or p8, is involved in multiple aspects of cancer, including DNA repair, transcription regulation, and the cell cycle, and its expression responds to stress signals induced by genotoxic signals and agents (**Martin et al., 2021**). NUPR1 influences cancer cell resistance (**Hamidi et al., 2012**) and promotes the proliferation of cancer cells bypassing the G0/G1 check point (**Brannon-Peppas et al., 2007**). In breast cancer cells, NUPR1 upregulates p21 transcription, allowing breast cancer cells to progress through the cell cycle, and it confers resistance to chemotherapeutic agents such as taxol and DOX (**Clark et al., 2008**; **Vincent et al., 2012**). Increased expression of NUPR1 has previously been associated with poor patient outcomes in certain types of cancers (**Jung et al., 2012**; **Mu et al., 2018**).

Histone deacetylase 11 (HDAC11) is the most recently discovered member of the HDAC family and the only member of class IV. It displays different expression levels and biological functions in different

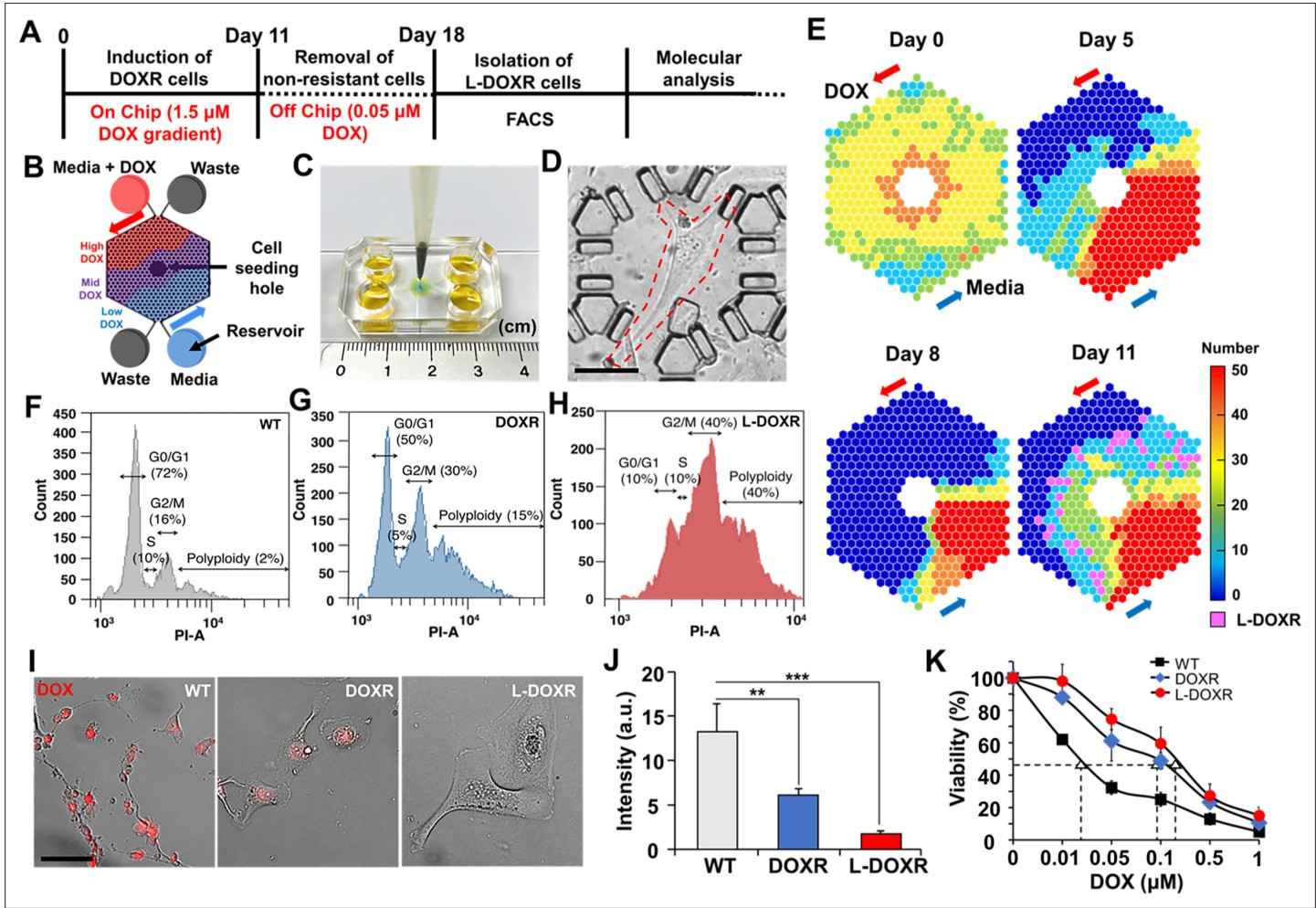

**Figure 2.** Tracking DOXR and L-DOXR cells induced by a DOX concentration-gradient in the CDRA chip and their cell cycle and drug resistance. (**A**) Experimental design. (**B**) Schematic of the chip. (**C**) Image of the CDRA chip. (**D**) L-DOXR cells (red dotted line) induced in the CDRA chip. (**E**) Tracking the number of live cells in each chamber of the chip for 11 days. L-DOXR cells are observed in some of the pink chambers on day 11. FACS analysis was used to assess the cell cycle of (**F**) WT cells, (**G**) DOXR cells, and (**H**) L-DOXR cells. (**I**) Red fluorescent intensity of WT cells, DOXR cells, and L-DOXR cells. Scale bar = 100 µm. (**J**) DOX efflux ability of WT cells, DOXR cells, and L-DOXR cells. **p<0.01, ***p<0.001, two-tailed Student's *t*-test. (**K**) DOX sensitivity of WT cells (The half-maximal inhibitory concentration (IC$_{50}$)=25 nM), DOXR cells (IC$_{50}$=100 nM), and L-DOXR cells (IC$_{50}$=200 nM).

human organs and systems. Its overexpression in various cancers, including hepatocellular, ovarian, myeloma, lymphoma, and breast cancers (*Gong et al., 2019*; *Huang et al., 2018*; *Liu et al., 2020*; *Yue et al., 2020*; *Zhou et al., 2018*), has suggested HDAC11 is an epigenetic regulator in human cancers. However, HDAC11 expression is negatively correlated with high-risk uveal melanomas and gliomas (*Dali-Youcef et al., 2015*), and HDAC11 knockout mice demonstrate increased tumor growth (*Sahakian et al., 2015*), indicating that its regulation of different cancer types is complex. Therefore, the pathophysiological roles of HDAC11 in various cancers should be evaluated.

## Results

### Formation and characterization of DOX surviving cells

Approximately 30 wild type MDA-MB-231 cells per microchamber were seeded through the cell seeding hole in the CDRA chip (*Figure 2A–C*). The day after seeding, the cells were perfused with gradients of medium and DOX (1.5 µM; *Figure 2A, B and E*). Cells exposed to a high concentration of DOX (high-DOX region) were killed within 5 days, whereas those exposed to an intermediate concentration of DOX (mid-DOX region) began to die on day 5 (*Figure 2E*). On day 8, DOXR cells appeared and proliferated in the mid-DOX region. On day 11, a population of phenotypically large cells (L-DOXR) appeared in the mid-DOX region (*Figure 2D and E*), suggesting that they emerge from stressful but tolerable conditions on the chip in areas where an intermediate concentration of DOX is perfused.

Cells were collected from the chip on day 12 and incubated with medium containing DOX (0.05 µM) for 7 days in 24 wells to remove non-resistant cells that might have originated from the low-DOX region (*Figure 2A and B*). Then, the DOXR cells were separated from the L-DOXR cells using FACS. The FACS cell cycle analysis showed that the proportions of polyploidy (cells greater than 4N+) in the WT cells, DOXR cells, and L-DOXR cells were 2, 15, and 40%, respectively (*Figure 2F–H*). The L-DOXR cells showed lower susceptibility to DOX than the WT and DOXR cells (*Figure 2I–K*). Taken together, these results suggest that the CDRA chip can rapidly induce the development of DOXR cells as well as a distinct population of L-DOXR cells.

### L-DOXR cells accelerate cancerous growth and tumor progression in TNBC

To better define the oncogenic properties of L-DOXR cells, including their potential role in chemoresistance in TNBC, we investigated their impact on cancer progression. Our results showed that L-DOXR cells exhibited significantly higher rates of proliferation and a greater proportion of Ki67-positive cells compared to WT cells (*Figure 3A and B*). An in vitro wound-healing assay showed L-DOXR cells migrated faster than WT cells, suggesting that the development of L-DOXR cells could increase the migration capacity of a TNBC cancer cell population (*Figure 3C*).

To ascertain whether the L-DOXR cells augmented tumorigenicity and conferred DOX-resistance in vivo, we generated an animal model of TNBC by subcutaneously injecting mice with either WT cells or L-DOXR cells and treating the tumor-bearing mice with either vehicle or DOX (*Figure 3D*). Irrespective of DOX treatment, the mice injected with L-DOXR cells showed much larger tumors compared to the mice injected with WT cells (*Figure 3E*). The tumor volume of L-DOXR cells treated with DOX and vehicle did not differ significantly (p>0.05), but the tumor volume of WT cells treated with DOX was significantly smaller than that of WT cells treated with vehicle (p<0.001) (*Figure 3—figure supplement 1A*). Our findings are consistent with hematoxylin and eosin (H&E) staining (*Figure 3G*, top) and immunohistochemical staining for proliferating cell nuclear antigen (PCNA) (*Figure 3G*, bottom) in the tumor tissues, which indicate that L-DOXR tumors did not exhibit a reduction in cell density or proliferation upon DOX treatment, in contrast to WT cells. Therefore, the L-DOXR cells in TNBC developed in the CDRA chip significantly enhanced carcinogenesis, and tumors initiated with L-DOXR cells were no longer sensitive to DOX.

L-DOXR cells exhibit increased genomic content (4N+) as compared to WT cells. The presence of cells with increased nuclear size and increased genomic content has been demonstrated to be associated with poor clinical outcomes in several types of cancers (*Alharbi et al., 2018*; *Amend et al., 2019*; *Fei et al., 2015*; *Imai et al., 1999*; *Liu et al., 2018*; *Lv et al., 2014*; *Mukherjee et al., 2022*; *O'Connor et al., 2002*; *Saini et al., 2022*; *Trabzonlu et al., 2023*). We analyzed the occurrence of

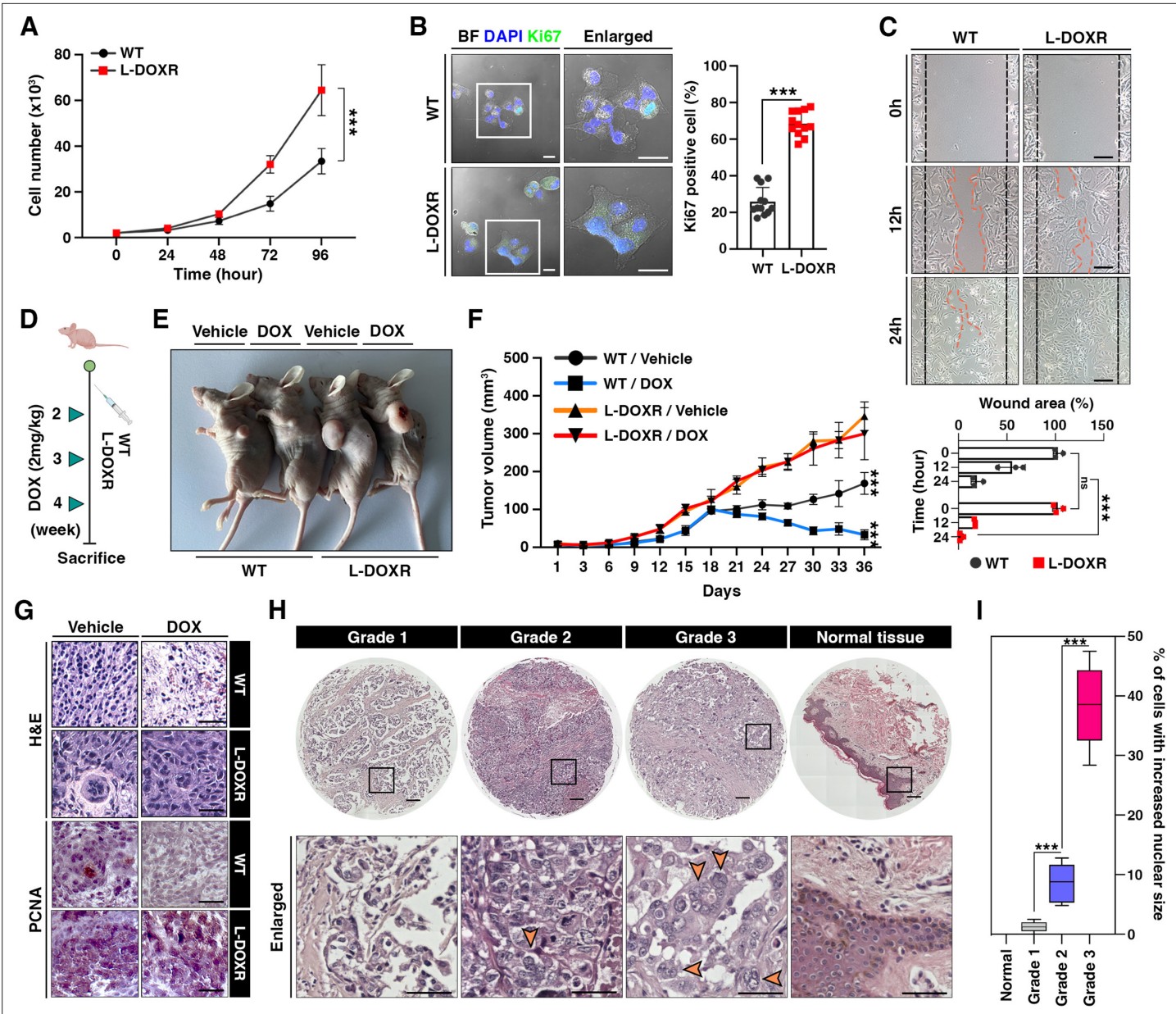

**Figure 3.** L-DOXR cells promote cancer growth and tumor progression in TNBC. (**A**) Cell proliferation assay of WT and L-DOXR cells by cell counting. (**B**) Ki67 immunofluorescence staining and intensity measurement in eight randomly selected fields to evaluate proliferative ability. Scale bars: 20 μm. (**C**) Wound healing assay to measure cell migration. The gap between cells was measured and shown as a bar graph (bottom). Scale bars: 50 μm. (**D, E**) Timeline showing subcutaneous injection of $1 \times 10^7$ WT cells and L-DOXR cells followed by DOX injection (2 mg/kg) once a week when tumor volume reached 150 mm³ (n=6 per group). A timeline demonstrating the subcutaneous injection of $1 \times 10^7$ WT cells and L-DOXR cells, followed by injection of DOX (2 mg/kg) into the tail vein (n=6 per group) once a week when the tumor volume reached 150 mm³. Representative tumors shown in photographs. (**F**) Tumor size measured with calipers every three days for up to 36 days. (**G**) Representative images of hematoxylin and eosin (H&E) staining (upper) and immunohistochemical staining for PCNA on paraffin sections of tumor tissues (bottom). Scale bars: 50 μm. (**H, I**) H&E staining of a TNBC tissue microarray with different tumor grades (grades 1, 2, 3, and negative) to detect L-DOXR cells. The number of L-DOXR cells was counted and analyzed from five randomly selected fields on each slide. The black boxes are magnified, and the orange arrows indicate L-DOXR cells. Scale bars: 500 μm. Data presented as mean ± SEM; ***p<0.001; Student's two-tailed, unpaired *t*-test (**A, B**); one-way ANOVA with Bonferroni's post-test (**C, F, I**).

The online version of this article includes the following figure supplement(s) for figure 3:

**Figure supplement 1.** L-DOXR accelerated cancerous growth and tumor progression in TNBC.

cells with increased nuclear size in human TNBC patients. A tissue microarray (TMA; n=130) found cells with increased nuclear size/genomic content only in TNBC patient tissues but not in normal breast tissue (*Figure 3—figure supplement 1B*). In addition, the number of cells with large nuclei in each tissue correlated with tumor grade (*Figure 3I*). Therefore, the presence of cells with increased genomic content in TNBC may indicate the presence of cells that are resistant to therapy.

## NUPR1 is a key mediator of chemoresistance

To elucidate the mechanism underlying the chemoresistance and oncogenic capacity of resistant cells, we performed an RNA sequencing (RNA-seq)-based transcriptome analysis to identify genes differentially expressed between WT and L-DOXR cells. Among the genes whose expression was significantly altered (fold change cut-off=2), 1212 were upregulated and 1,602 were downregulated in the L-DOXR cells (*Figure 4A*). A DAVID gene ontology term analysis of genes upregulated in the L-DOXR cells (false discovery rate <0.05) indicated that genes involved in cancer progression were most represented. An Ingenuity Pathway Analysis (IPA) revealed that *NUPR1*, whose upregulation is associated with malignancy of cancer and the chemoresistance network (*Wang et al., 2021*), was top-ranked, and antioxidant signaling was the most enriched pathway along with other cancer-promoting signaling such as tumor necrosis factor receptor 2, mitogen-activated protein kinase, and phospholipase signaling (*Figure 4B*). Notably, the upstream regulator analysis in IPA revealed that *NUPR1* is a high-rank regulator and is responsible for 4.4% (53/1212) of the genes actively transcribed in the L-DOXR cells (cut-off=1.5, p<0.05) (*Figure 4C*).

The clinical relevance of *NUPR1* expression in TNBC was investigated using a cohort of patients treated with chemotherapy by performing a meta-analysis of all the datasets in Kaplan-Meier plotter (https://kmplot.com/analysis/index.php?p=service&cancer=breast; *Lánczky and Győrffy, 2021*). The overall survival rate was significantly lower in patients with high *NUPR1* mRNA expression than in patients whose *NUPR1* mRNA expression was low (high, n=34; low, n=32; p=0.037; *Figure 4D*). Similarly, in other datasets GSE12093 (*Zhang et al., 2009*; *Figure 4E*) and GSE16391 (*Desmedt et al., 2009*), chemotherapy-treated breast cancer patients with significantly lower survival rates expressed higher level of *NUPR1* (p=0.027 and 0.0003, respectively; *Figure 4—figure supplement 1A*), suggesting that high *NUPR1* expression is associated with poor clinical outcomes among TNBC patients.

Consistent with the RNA-seq analysis of L-DOXR cells, increased expression of *NUPR1* in both the L-DOXR cells and L-DOXR cell-derived xenografts were observed in reverse transcriptase-quantitative polymerase chain reaction (RT-qPCR; *Figure 4F*). However, in contrast to L-DOXR cells, mRNA level of *NUPR1* was barely detectable in the WT cells and WT cell-derived tumor tissues. While DOXR cells exhibited a marked increase in *NUPR1* expression compared to the WT cells, this expression was substantially less than that observed in L-DOXR cells, as detailed in *Figure 4—figure supplement 1B*. Furthermore, transactivation activity of the *NUPR1* promoter was highly elevated in L-DOXR cells but not in WT cells (*Figure 4G*). These results indicate that *NUPR1* expression is highly enhanced in L-DOXR cells. Silencing *NUPR1* expression abolished the cell viability of DOX-treated L-DOXR cells, but it did not decrease the cell viability of vehicle-treated L-DOXR cells, suggesting that *NUPR1* depletion could eliminate DOX resistance in L-DOXR cells (*Figure 4—figure supplement 1C*). Regardless of DOX-treatment, *NUPR1* depletion did not affect the chemosensitivity of WT cells. In addition, we showed that overexpression of *NUPR1* in the WT cells attenuates DOX-induced cytotoxicity (*Figure 4I*). These results suggest that *NUPR1* upregulation may be a major driver of chemoresistance in L-DOXR cells.

To define the potential role of NUPR1 in mediating chemoresistance in TNBC, we treated WT cells and L-DOXR cells with ZZW-115, a NUPR1 inhibitor that alters its nuclear localization (*Lan et al., 2020*), in the absence or presence of DOX. ZZW-115 treatment led to re-sensitization of L-DOXR cells to DOX in a dose-dependent manner, whereas the WT cells barely responded to ZZW-115 (*Figure 4J*). Delocalization of NUPR1 and increased cell death caused by ZZW-115 were confirmed by immunocytochemistry (*Figure 4—figure supplement 1D*) and active CASPASE-3 and poly (ADP-ribose) polymerase (PARP) cleavage (*Figure 4K*). To further verify whether NUPR1 inhibition could overcome DOX resistance and enhance drug response in L-DOXR cells, we treated xenograft model mice with DOX and two doses of ZZW-115 (*Figure 4L–P*). The addition of ZZW-115 to DOX in the xenograft models resulted in a reduction of tumor volume compared to DOX-only-treated tumors (–469.5 ± 25.20 mm$^3$ [2.5 mg/kg] and –627.2±15.36 [5.0 mg/kg]) (*Figure 4M-O*; *Figure 4—figure supplement 1E*) and

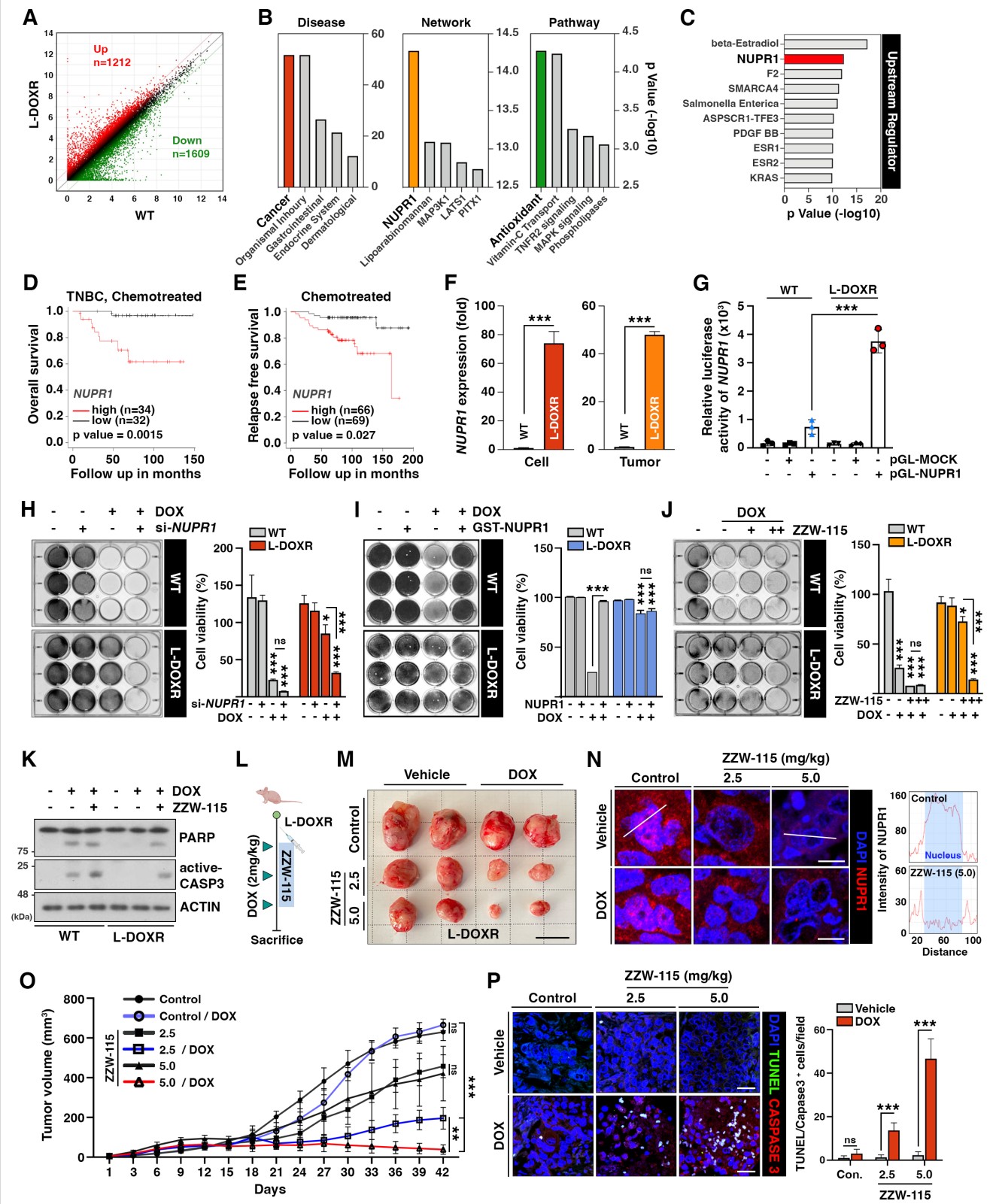

**Figure 4.** NUPR1 is a key mediator of chemoresistance in L-DOXR cells. (**A**) Volcano plot of differential gene expression between WT and L-DOXR cells. Cut-off criteria included a fold change of 2. (**B, C**) Ingenuity Pathway Analysis (IPA) of the RNA-sequencing data shows disease and disorders (left), causal network (middle), canonical pathways (*Yue et al., 2020*), and upstream regulator (**C**). The top five ranks are presented. Cut-off criteria are p<0.05 and a false discovery rate (FDR) *q*-value <0.05. (**D**) Kaplan-Meier (*Dai et al., 2013*) survival curve represents the overall survival rate in chemotherapy-treated

*Figure 4 continued on next page*

*Figure 4 continued*

TNBC patients (n=66) based on low vs. high *NUPR1* expression from the meta-analysis in KM plotter. (**E**) KM survival curve representing the relapse-free survival rate in chemotherapy-treated patients (n=135) based on low vs. high *NUPR1* expression from GSE12093. (**F**) Reverse transcriptase-quantitative polymerase chain reaction (RT-qPCR) analysis of *NUPR1* mRNA expression in cells (left) and tumor tissue from a mouse xenograft (*Yue et al., 2020*). The values were normalized to the level of the control (*Jazaeri et al., 2005*). (**G**) The relative luciferase activity of the *NUPR1* promoter was measured in WT cells and L-DOXR cells. (**H**) Cell viability was measured among si-*NUPR1* transfected cells treated with DOX using a crystal violet staining assay. (**I**) Cell viability was measured using GST-NUPR1 transfected cells with DOX. (**J**) Cell viability was measured after administering DOX with and without ZZW-115 (*NUPR1* inhibitor). The bar graph indicates the average density of dyed crystal violet. (**K**) Apoptotic proteins were detected by immunoblotting from WT cells and L-DOXR cells with and without DOX/ZZW-115. (**L**) Timeline demonstrating the subcutaneous injection of $1 \times 10^7$ L-DOXR cells followed by injections of doxorubicin (4 mg/kg) or ZZW-115 (2.5 mg/kg, 5.0 mg/kg) into the tail vein (n=6 per group). (**M**) The photographs show representative tumors. Scale bar: 2 cm. (**N**) Representative images showing immunohistochemical staining for NUPR1 in PCGGs with and without ZZW-115 and DOX treatment (left). Localization of *NUPR1* in the control and ZZW-115 (5.0 mg/kg)-injected tumors was analyzed by ImageJ (*Yue et al., 2020*). Scale bars: 20 µm. (**O**) Animals were monitored for up to 42 days, and tumor size was measured using calipers at three-day intervals. (**P**) Representative images showing immunohistochemical staining for TUNEL and active-caspase 3 on paraffin sections of tumor tissues. Scale bars: 20 µm. All data are presented as mean ± SEM; *p<0.05, **p<0.01, ***p<0.001; Student's two-tailed, unpaired *t*-test (**F**); one-way ANOVA with Bonferroni's post-test (**G, H, I, J, O**).

The online version of this article includes the following source data and figure supplement(s) for figure 4:

**Source data 1.** Original image for the western blot analysis in *Figure 4K*.

**Figure supplement 1.** NUPR1 is a key mediator of chemoresistance in L-DOXR.

**Figure supplement 1—source data 1.** Original image of the RNA expression in *Figure 4—figure supplement 1C*.

induced significant cell death (*Figure 4P*). These findings suggest that NUPR1 inhibition can over-come chemoresistance in highly aggressive L-DOXR cell-induced tumors in xenograft model mice.

## HDAC11 suppression leads to *NUPR1* upregulation

To gain insights into the molecular mechanism underlying *NUPR1* upregulation in L-DOXR cells, we aimed to identify a potent regulator of its gene expression. Because epigenetic alterations affect gene expression and are usually associated with cancer progression (*Baxter et al., 2014*), we first examined the DNA methylation status of the *NUPR1* promoter region. However, we did not find any remarkable changes in promoter methylation between WT cells L-DOXR cells (*Figure 5—figure supplement 1A*). Intriguingly, chromatin immunoprecipitation (ChIP)-qPCR using the histone H3 at lysine 27 (H3K27)-acetylation antibody revealed H3K27 acetylation in L-DOXR cells, specifically in promoter region 3 (*Figure 5A*). Spurred by our finding of enriched acetylation in L-DOXR cells, we attempted to identify a putative epigenetic regulator, such as a histone acetyltransferase or HDAC, that could be involved in the increased acetylation of *NUPR1*. An RNA-seq analysis of HATs and HDACs in WT cells and L-DOXR cells showed almost no detectable mRNA expression of *HDAC11* in L-DOXR cells (*Figure 5—figure supplement 1B*), which we confirmed by RT-qPCR (*Figure 5C*). *HDAC11* expression was also dramatically reduced in tumors from L-DOXR cell-derived xenografts compared with tumors derived from WT cells (*Figure 5D*). In addition, the protein expression of NUPR1 and HDAC11 was inversely correlated in L-DOXR cells and WT cells (*Figure 5—figure supplement 1C, D*), suggesting that low levels of HDAC11 in L-DOXR cells might contribute to the upregulation of *NUPR1* through enriched acetylation in its promoter region. Indeed, forced expression of HDAC11 elicited a dramatic reduc-tion in H3K27 acetylation in the L-DOXR cells promoter region (*Figure 5F*), which reduced the mRNA expression of *NUPR1* in a dose-dependent manner not seen in the parental WT cells (*Figure 5E*) and also greatly impaired the promoter activity in the L-DOXR cells (*Figure 5G*). Moreover, HDAC11 inhib-itor treatment in WT cells augmented the expression of *NUPR1*, presumably, reflecting the reverting of promoter acetylation (*Figure 5H*). These data clearly demonstrate that HDAC11 mediates *NUPR1* promoter deacetylation, underscoring that the suppressed expression of HDAC11 in L-DOXR cells allows *NUPR1* to escape deacetylation and thereby causes its aberrant high expression.

In a tissue microarray (TMA) of TNBC patient tissues (n=130), we verified that, as tumor grade increased, NUPR1 expression increased and HDAC11 expression decreased (*Figure 5I*). In addition, a KM plot analysis of breast cancer patients (n=500, HER negative) from GSE25066 (*Hatzis et al., 2011*) showed that patients with low *HDAC11* expression had significantly shorter survival times than patients with high *HDAC11* expression after chemotherapy (*Figure 5—figure supplement 1E*). Thus, these data emphasize that *NUPR1* is inversely correlated with *HDAC11* level in TNBC patients, and

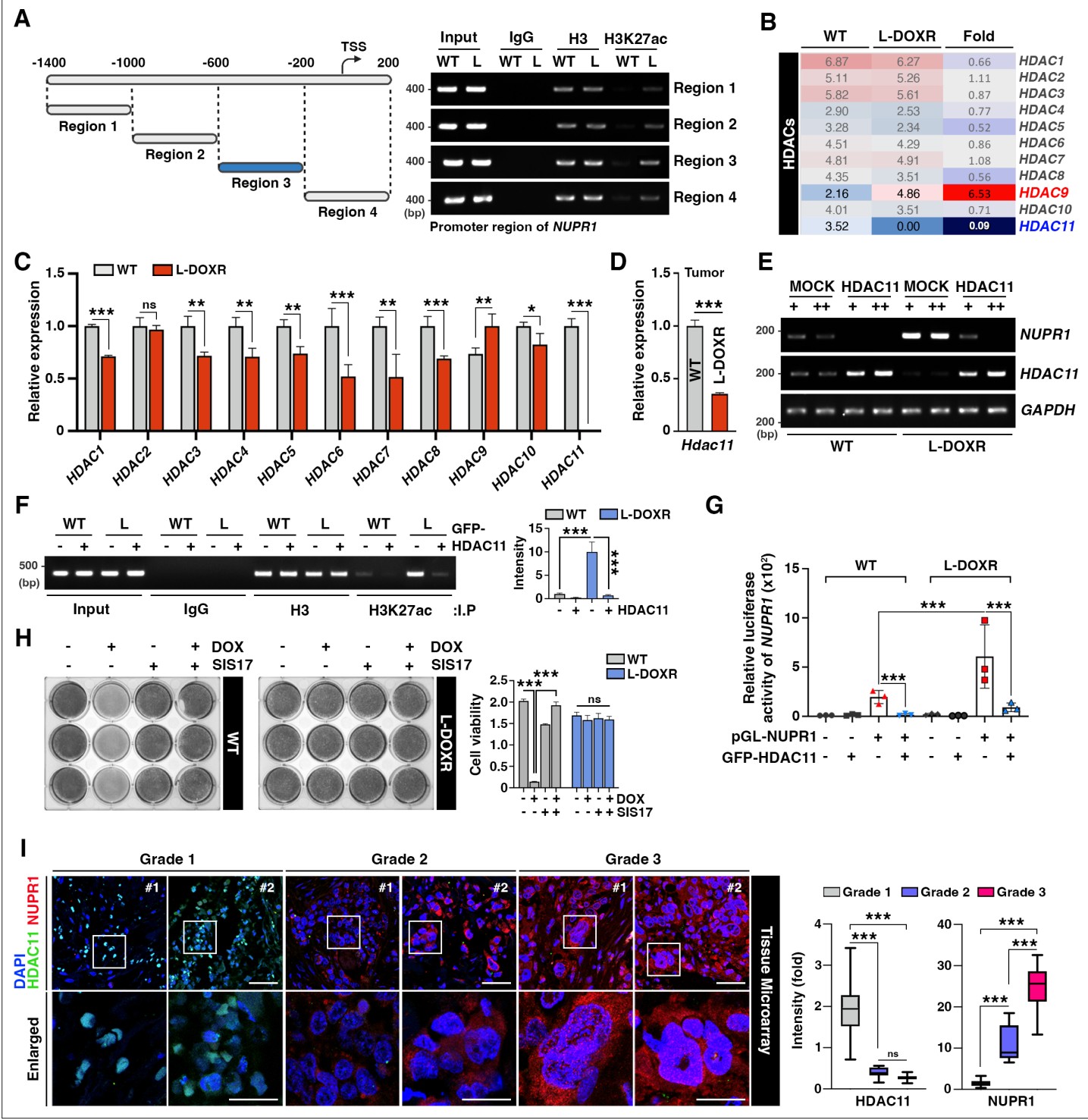

**Figure 5.** HDAC11 suppression leads to *NUPR1* upregulation in L-DOXR cells. (**A**) Schematic diagram showing the promoter region of *NUPR1*. A ChIP assay was performed with qPCR on WT cells and L-DOXR cells using anti-H3 and H3K27ac antibodies. L, L-DOXR cells. (**B**) A heat map representing the relative mRNA expression levels of *HDACs* in WT cells and L-DOXR cells. (**C**) Real-time PCR analysis of the mRNA expression of the indicated genes in WT cells and L-DOXR cells. (**D**) The mRNA expression of *HDAC11* in L-DOXR cells-derived tumor tissue was measured by RT-qPCR. (**E**) The mRNA expression of *NUPR1* and *HDAC11* was measured in cells transfected with either GFP-MOCK or HDAC11. (**F**) A ChIP assay was performed after transfecting WT and L-DOXR cells with GFP-MOCK or HDAC11 using anti-H3 or H3K27ac antibodies. Acetylated-histone levels were determined by RT-qPCR with specific primers (−600/−200). L, L-DOXR cells. (**G**) The relative luciferase activity of the *NUPR1* promoter was measured after transfecting WT cells and L-DOXR cells with GFP-HDAC11. (**H**) Cell viability was measured among SIS17-treated cells with DOX using a crystal violet staining assay.

*Figure 5 continued on next page*

*Figure 5 continued*

(I) Representative images show the expression of NUPR1 (*Gao et al., 2002*) and HDAC11 (*Liedtke et al., 2008*) on a TNBC TMA with different tumor grades (grades 1, 2, and 3). Quantitative analysis of the intensity of NUPR1 and HDAC11 is displayed (*Yue et al., 2020*). White boxes are magnified. Scale bars: 50 μm (upper) and 25 μm (bottom). All data are presented as means ± SEM; *p<0.05, **p<0.01, ***p<0.001; Student's two-tailed, unpaired *t*-testing (**C, D**); one-way ANOVA with Bonferroni's post-test (**G, I**).

The online version of this article includes the following source data and figure supplement(s) for figure 5:

**Source data 1.** Original image for the promoter region in *Figure 5A*.

**Source data 2.** Original image for the RNA expression in *Figure 5E*.

**Source data 3.** Original image for the RNA expression in *Figure 5F*.

**Figure supplement 1.** HDAC11 suppression leads to *NUPR1* upregulation in L-DOXR.

**Figure supplement 1—source data 1.** Original image of the RNA expression in *Figure 5—figure supplement 1A*.

**Figure supplement 1—source data 2.** Original image of the western blot in *Figure 5—figure supplement 1C*.

that the epigenetic dysregulation of *NUPR1* caused by low HDAC11 level may cause the chemoresistance that dictates the development of L-DOXR cells in TNBC.

## Discussion

TNBC is the most aggressive subtype of breast cancer, and chemotherapy is a mainstay of treatment. However, chemoresistance is common and contributes to the long-term survival of TNBC patients (*Liedtke et al., 2008*). In this study, we obtained DOX-resistant cells that exhibit an enlarged phenotype with increased genomic content. We also identified a mechanism for that drug resistance through epigenetic control of the NUPR1/HDAC11 axis in TNBC. L-DOXR cells and L-DOXR cell-derived tumor tissues showed high-level expression of *NUPR1*, which was consistent with the poor clinical outcomes, including low overall survival (OS) and disease-free survival (DFS), in chemotherapy-treated TNBC patients with high *NUPR1*. Our findings demonstrated that *NUPR1* expression in L-DOXR cells is induced by acetylation of the *NUPR1* promoter through the aberrantly restricted expression of HDAC11. The identification of *NUPR1* as a novel epigenetic target of HDAC11 in L-DOXR cells helps to explain how L-DOXR cells acquire chemoresistance. HDAC11 is the most recently discovered HDAC, and its pathophysiological role is poorly understood. For example, HDAC11 has a positive correlation with tumor growth, but its incongruously high expression also conferred longer DFS and OS in pancreatic tumor patients (*Klieser et al., 2017*). HDAC11 is overexpressed in certain cancer cell lines, including prostatic (PC-3) (*Huo et al., 2020*), ovarian (SK-OV-3) (*Zhou et al., 2018*), and breast cancer (MCF-7) (*Gao et al., 2002*) cells, and HDAC11 inhibition has shown beneficial effects in neuroblastoma cells (*Thole et al., 2017*) and Hodgkin lymphoma (*Buglio et al., 2011*). However, HDAC11 expression is inversely correlated with high-risk uveal melanomas and gliomas (*Dali-Youcef et al., 2015*), and HDAC11 knockout mice have increased lymphoma tumor growth (*Sahakian et al., 2015*). HDAC11 inhibition promotes breast cancer cell metastasis (*Leslie et al., 2019*). In basal-like breast cancer cells with decreased HDAC11 expression, overexpression of HDAC11 did not inhibit tumor growth but did inhibit invasion and metastasis (*Denkert et al., 2017*). In addition, the Cancer Genome Atlas shows that *HDAC*11 promoter methylation is associated with a poor prognosis of ovarian cancer patients (*Dai et al., 2013*), suggesting the need for in-depth studies of the specific mechanisms of HDAC11 in specific tumors. In this study, we observed extremely low HDAC11 expression in L-DOXR cells compared with WT cells, and we confirmed that its expression is much lower in patients with high-grade TNBC tumors than in those with low-grade tumors. We also found a positive correlation between its expression and disease-free survival (*Figure 5—figure supplement 1E*). Because we identified that NUPR1 as a novel target of HDAC11, and drastically decreasing the expression of HDAC11 causes aberrantly high expression of NUPR1 in L-DOXR cells and TNBC patients (*Figure 5H*), it is plausible that limited expression of HDAC11 leads to a high NUPR1 level to acquire chemoresistance. It is also possible that HDAC11 expression may be suppressed in chemoresistant TNBC cells by a specific regulator that requires further elucidation.

In breast cancer, aberrations in histone modification such as acetylation have been shown to be important for tumor progression and have been proposed as a promising therapeutic target (*Cheng et al., 2019*). HDACs have been an attractive therapeutic strategy for restoring both acetylation and

gene expression, with the potential benefit of being better tolerated than cytotoxic chemotherapy. Epigenetic modulation has also been hypothesized as a mechanism of chemoresistance. In this study, we showed that NUPR1 overexpression upon acquisition of DOX resistance leads to upregulation of cancer-promoting signaling. Moreover, we demonstrated that NUPR1 inhibition with ZZW-115 reconstitutes the drug sensitivity of L-DOXR cells and HDAC11 overexpression inhibited *NUPR1* expression by eliciting deacetylation of the *NUPR1* promoter region in L-DOXR cells. Thus, despite the promising anti-tumor effects of HDAC inhibitors (HDACi) in preclinical models, our results suggest the importance of evaluating HDACi as therapeutic candidates in the context of drug-resistance in TNBC.

The L-DOXR cells observed in our study resemble the previously reported polyaneuploid cancer cell (PACC) state (*Chen et al., 2019*; *Zhang et al., 2014a*). Cells in the PACC state (PACCs) have been described by many names including polyploid giant cancer cells (PGCCs) and are present in multiple high-grade and post-treatment cancers (*Chen et al., 2019*; *Zhang et al., 2014a*). Various environmental factors, including hypoxia (*Zhang et al., 2014a*), anticancer drugs (*Islam et al., 2018*; *Jia et al., 2012*; *Zhang et al., 2014b*), and radiation therapy (*Zhang et al., 2021*) have all been reported to lead to induction of the PACC state (*Zhang et al., 2014b*; *Ahn et al., 2004*). Cells in the PACC state demonstrate plasticity and have the capacity to further divide and produce progeny, contributing to an increase in tumor heterogeneity and therapeutic resistance (*Niu et al., 2016*). The mechanism by which the PACC state confers drug resistance is unknown.

Our results demonstrate that clinically meaningful resistant cells can be obtained within a few weeks using the CDRA chip to mimic the spatiotemporally heterogeneous ecosystem of cancer cells in the tumor tissues of patients receiving chemotherapy. Although large cells with high genomic content are often found in cancer patient tissues, their isolation is technically difficult, which is an obstacle to studying how they contribute to chemoresistance in cancer patients. Therefore, our methodology, examining the expression of genes involved in the chemoresistance of chip-derived large cells and comparing those results with gene expression data from patient tissues in which cells with high genomic content are found, opens a new avenue for understanding the mechanism of chemoresistance. Because the chip requires approximately 15,000 cells each, it can be also used to predict resistance in patients prior to chemotherapy (*Garraway and Jänne, 2012*).

## Materials and methods
### Fabrication of the CDRA chip
The CDRA chip was fabricated using soft lithography, as previously described (*Han et al., 2016*; *Han et al., 2019*). The chip contained a patterned array of 444 hexagonal microchambers, each with a diameter of 200 µm. In the outermost chambers, 5-µm-wide channels allowed medium with and without DOX to perfuse into the interior microchambers. Each interior microchamber had three gates through which the cells could move into the connected chambers.

### Cell culture
The MDA-MB-231 TNBC cell line was purchased from ATCC (Manassas, VA, USA) and cultured in RPMI-1640 medium (HyClone, Logan, UT, USA) supplemented with 10% fetal bovine serum (HyClone), 100 units per mL of penicillin (Life Technologies, Carlsbad, CA, USA), and 100 µg/mL of streptomycin (Life Technologies) and maintained at 37°C with 5% $CO_2$.

### Operation of the CDRA chip
The chip was prepared before cell seeding as described before (*Han et al., 2016*). A total of $1 \times 10^5$ cells/10 µL was suspended in culture medium, and 1 µL of the solution was gently added to the chip using a pipette with a tip through the cell seeding hole. The hole was plugged with a sterilized stainless pin, and the chip was incubated at 37°C with 5% $CO_2$ overnight. The next day, 250 µL of culture medium and culture medium containing 1.5 µM DOX were added to two of the diagonal reservoirs, and 50 µL of culture medium was added to the rest of the diagonal reservoirs. The fresh culture medium and drug were replaced every day. After 11 days, trypsin (Gibco) was added to the chip, which was incubated at 37°C for 5 min. The detached cells were flushed out of the chip and collected from the reservoirs by injecting 1 mL of culture medium through the seeding hole with a needle-free

syringe. To remove non-resistant cells, the collected cells were grown in medium containing 0.05 μM DOX for 1 week (*Figure 2a*).

## L-DOXR isolation using FACS

DOXR cells were seeded in a 10 mm cell culture dish for 1 day and then stained with 5 μg/mL of Hoechst-33342 at 37°C for 5 min and analyzed on a FACSAria Fusion (BD Biosciences, Franklin Lakes, NJ, USA).

## Cell cycle analysis using FACS

Cells were collected in a 15 mL tube and fixed in pre-cooled 70% ethanol at 4°C for 1 hr. The cells were permeabilized in 0.25% Triton X-100 with phosphate buffered saline (PBS, pH 7.4) at 4°C for 15 min and then stained with 20 μg/mL of propidium iodide (Sigma-Aldrich) containing 10 μg/mL of ribonuclease A at room temperature for 30 min. The stained cells were analyzed in the FACSAria Fusion.

## DOX efflux

About $1 \times 10^5$ cells were incubated in a 6-well plate (Corning Inc) containing RPMI-1640 medium with 5 μM (final concentration) DOX at 37°C for 3 hr, and then the medium was replaced with fresh RPMI-1640 without DOX. After 24 hr, fluorescent images were captured using a DeltaVision Elite microscope (GE Healthcare, Chicago, IL, USA). Then, 10 cells were randomly chosen from the images, and their fluorescence intensity at 585 nm was analyzed using ImageJ (NIH, Bethesda, MD, USA).

## Cell viability

To assess the cytotoxic effects of DOX on cells, approximately $10^3$ cells were incubated in a 96-well plate with DOX (0–1 μM) for 72 hr at 37°C. Their viability was measured using EZ-Cytox reagent (Daeillab Service, Seoul, Korea). The percentage of viable cells was calculated by dividing the number of viable cells at each DOX concentration by the number of cells cultured without DOX.

## RNA sequencing

Total RNA from untreated and treated MDA-MB-231 cells was extracted using a RNeasy Mini Kit (Qiagen, Germantown, MD, USA). RNA sequencing was performed on the NextSeq 500 sequencing platform (Illumina, San Diego, CA, USA). Biological functions were determined using IPA web-based bioinformatics software (QIAGEN). A twofold change in treated cell gene expression was used as the cut-off value indicating a significant change in expression compared with that in untreated MDA-MB-231 cells.

## Antibodies and chemicals

Anti-PCNA (ab29), Ki67 (ab15580), NUPR1 (ab234696), and active-Caspase 3 (ab2302) antibodies were acquired from Abcam (Cambridge, UK). Anti-HDAC11 (H4539) and HDAC11 (WH0079885M1) antibodies were acquired from Sigma-Aldrich. Anti-PARP (9542 S) antibody was obtained from Cell Signaling Technology (Danvers, MA, USA). Anti-ACTIN (sc-47778) antibody was obtained from Santa Cruz Biotechnology (Dallas, TX, USA). Dimethyl sulfoxide (D2447), ZZW-115 (HY-111838A), and DOX (D1515) were acquired from Sigma-Aldrich.

## Tissue microarray and immunohistochemistry

Slides of TNBC and normal tissues were obtained from US Biomax (BR1301) (Derwood, MD, USA) consisting of 125 cases of TNBC specimens, whose characteristics, including pathology grade, TNM, clinical stage, and IHC (ER, PR, HER2) results are available online (BR1301 Tissue Array and Tissue Microarray of premade types). For staining, each slide was deparaffinized and permeabilized using 0.25% Triton X-100 in PBS for 2 h. The slides were immunostained using primary antibodies and incubated overnight at 4 °C and then incubated for 1 hr at room temperature with secondary antibodies (Alexa Fluor-488 or –546). Nuclei were counterstained with 4',6-diamidino-2-phenylindole. Z-stacked images of the stained tissues were acquired using a ZEISS LSM 710 confocal microscope (Zeiss, Oberkochen, Germany).

## Western blot

Transfected cells were washed with PBS and treated with ice-cold lysis buffer (50 mM Tris-Cl, pH7.4; 150 mM NaCl; 1 mM EDTA; 0.5% Triton X-100; 1.5 mM $Na_3VO_4$; 50 mM sodium fluoride; 10 mM sodium pyrophosphate; 10 mM glycerophosphate; 1 mM phenylmethylsulfonyl fluoride, and protease cocktail (Calbiochem, San Diego, CA, USA)). Equal amounts of proteins were denatured, resolved on SDS-PAGE, and transferred to nitrocellulose membranes (Pall Life Science, Port Washington, NY, USA) (*Woo et al., 2022*).

## RT-qPCR

To compare the mRNA levels of WT and L-DOXR cells, RT-qPCR was performed. Total RNA was isolated from cells or tumors using a Mini BEST Universal RNA Extraction Kit (Takara, Shiga, Japan). cDNA was prepared from total RNA by reverse transcription using oligo-dT primers (Takara). RT-qPCR was conducted using SoFast EvaGreen Super Mix (Bio-Rad, Hercules, CA, USA) according to the manufacturer's instructions. glyceraldehyde 3-phosphate dehydrogenase (*Gapdh*) was used as an internal control for quantitation of target gene expression. A total reaction mixture with a volume of 20 μl was amplified in a 96-well PCR plate (Bio-Rad). The primer sets used are listed in *Supplementary file 1*.

## Luciferase assay

Cells were plated in culture plates and transfected with 100 ng of *NUPR1*-promoter-luciferase reporter and 30 ng of *Renilla* reporter vector in 6-well plates and then incubated for 24 hr (*Yu et al., 2022*). The cells were lysed, and luciferase assays were performed using a dual luciferase assay kit (Promega, Madison, WI, USA) according to the manufacturer's instructions. The transfection efficiency was normalized against *Renilla* luciferase activity, and the transfection of genes was confirmed using immunoblotting. All assays were performed at least in triplicate.

## ChIP assay

ChIP assays were performed using a ChIP Assay Kit (cat. 17–259; Millipore, Temecula, CA, USA) according to the manufacturer's instructions. Primers from multiple sites relative to the transcription start site were designed and pretested in both the input and ChIP samples. Purified DNA was subjected to qPCR with primers against the *NUPR1* promoter region. The primer sets used are listed in *Supplementary file 1*.

## Survival analysis

The KM plots were taken from https://kmplot.com/analysis/index.php?p=service&cancer=breast (*Desmedt et al., 2009*). We chose TNBC patients as follows: ER status IHC: ER-negative; ER status array: ER-negative; PR status IHC: PR negative; and HER2 status array: HER2 negative for meta-analysis and retrieved from the NCBI GEO database GSE12093 (*Zhang et al., 2009*), GSE16391 (*Desmedt et al., 2009*), and GSE25066 (*Hatzis et al., 2011*).

## Animal

All animal experiments were reviewed and approved by the Institutional Animal Care and Use Committee (IACUC) of Sungkyunkwan University School of Medicine (SUSM, SKKUIACUC2021-03-47-1). All experimental procedures were performed according to the regulations of the IACUC guidelines of Sungkyunkwan University.

## Xenograft

Procedures for the animal studies were described previously (*Hwang et al., 2016*). Briefly, 6- to 8-week-old female Balb/c nude mice (Orientbio Inc, Seongnam, Korea) were housed in laminar-flow cabinets under specific pathogen-free conditions. Approximately $1 \times 10^7$ cells of WT cells or treated cells were resuspended in 100 μL of a 1:1 ratio of PBS and Matrigel (Corning Inc, Corning, NY, USA, #354234) and subcutaneously injected into each mouse. The tumor size was monitored every three days using calipers, and the tumor volume (V) was calculated using the formula $V = (L \times W^2)/2$, where L was the length and W was the width of the tumor. When the tumor volume reached 150 mm$^3$, the

tail veins of the mice were injected with 2 mg/kg of DOX for *Figure 3D–G* or 2.5 mg/kg or 5.0 mg/kg of ZZW-115 (daily) with and without 2 mg/kg of DOX for *Figure 4K–O*.

## Statistical analysis

All statistical analyses were performed using Prism 8 (GraphPad Software, San Diego, CA, USA). In general, statistical analyses were performed using ANOVA and Student's *t*-test. Two-tailed and unpaired *t*-tests were used to compare two conditions. Two-way ANOVA with Tukey's post hoc test was used to analyze multiple groups. One-way ANOVA with Bonferroni's post hoc test was used for comparisons of ages and genotypes. Data are represented as mean ± standard error of the mean (SEM) unless otherwise noted, with asterisks indicating $*p<0.05$, $**p<0.01$, and $***p<0.001$.

## Acknowledgements

This work was equally supported by a grant of the Korea Dementia Research Project through the Korea Dementia Research Center (KDRC), funded by the Ministry of Health & Welfare and Ministry of Science and ICT, Republic of Korea (grant number: HU21C0157) to JYA and funded by Technology Innovation Program (or Industrial Strategic Technology Development Program-Development of disease models based on 3D microenvironmental platform mimicking multiple organs and evaluation of drug efficacy) (20008413) funded by the Ministry of Trade, Industry & Energy (MOTIE, Korea) to SP Additionally, WL was supported by the Fostering Global Talents for Innovative Growth Program, grant P0008746, overseen by the Korea Institute for Advancement of Technology (KIAT).

## Additional information

### Funding

| Funder | Grant reference number | Author |
| --- | --- | --- |
| Korea Dementia Research Center | HU21C0157 | Jee-Yin Ahn |
| Ministry of Trade, Industry and Energy | 20008413 | Sungsu Park |
| Korea Institute for Advancement of Technology | P0008746 | Wanyoung Lim |

The funders had no role in study design, data collection and interpretation, or the decision to submit the work for publication.

### Author contributions

Wanyoung Lim, Investigation, Writing – original draft; Inwoo Hwang, Formal analysis, Investigation, Visualization, Writing – original draft; Jiande Zhang, Investigation, Methodology; Zhenzhong Chen, Formal analysis; Jeonghun Han, Bon-Kyoung Koo, Data curation, Writing – review and editing; Jaehyung Jeon, Youngkwan Kim, Investigation; Sangmin Kim, Jeong Eon Lee, Resources, Data curation, Writing – review and editing; Kenneth J Pienta, Conceptualization, Data curation, Writing – original draft, Writing – review and editing; Sarah R Amend, Writing – original draft, Writing – review and editing; Robert H Austin, Conceptualization, Writing – original draft, Writing – review and editing; Jee-Yin Ahn, Resources, Data curation, Formal analysis, Funding acquisition, Investigation, Methodology, Writing – original draft, Writing – review and editing; Sungsu Park, Conceptualization, Data curation, Formal analysis, Supervision, Funding acquisition, Validation, Investigation, Writing – original draft, Project administration, Writing – review and editing

### Author ORCIDs

Inwoo Hwang 
Kenneth J Pienta 
Jee-Yin Ahn 
Sungsu Park 

### Ethics

All animal experiments were reviewed and approved by the Institutional Animal Care and Use Committee (IACUC) of Sungkyunkwan University School of Medicine (SUSM, SKKUIACUC2021-03-47-1). All experimental procedures were performed according to the regulations of the IACUC guidelines of Sungkyunkwan University.

Reviewer #1 (Public Review): https://doi.org/10.7554/eLife.88830.3.sa1
Reviewer #2 (Public Review): https://doi.org/10.7554/eLife.88830.3.sa2
Reviewer #3 (Public Review): https://doi.org/10.7554/eLife.88830.3.sa3
Author Response https://doi.org/10.7554/eLife.88830.3.sa4

## Additional files

### Supplementary files

• MDAR checklist

• Supplementary file 1. List of primer sequences for RT-qPCR and ChIP assay.

### Data availability

RNA-seq raw and processed data files have been uploaded to the Gene Expression Omnibus and can be accessed using the following accession code GSE256086 for transcriptional profile.

The following dataset was generated:

| Author(s) | Year | Dataset title | Dataset URL | Database and Identifier |
|---|---|---|---|---|
| Lim W, Hwang I, Zhang J, Chen Z, Han J, Jeon J, Koo B, Kim S, Lee J, Pienta K, Amend S, Austin R, Ahn J, Park S | 2024 | Exploration of Mechanisms of Drug Resistance in a Microfluidic Device and Patient Tissues | https://www.ncbi.nlm.nih.gov/geo/query/acc.cgi?acc=GSE256086 | NCBI Gene Expression Omnibus, GSE256086 |

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
