## [Editor Report · eLife assessment]

This study based on the use of Cancer Drug Resistance Accelerator (CDRA) chip is **valuable** as a platform technology to assess chemoresistance mechanisms. The strength is **convincing** from the technological point of view. However, the use of a single cell line model is a limitation. However we acknowledge the authors' plan to further validate their current findings across multiple TNBC cell lines.

---

## [Referee Report · Reviewer #1 (Public Review)]

Lim W et al. investigated the mechanisms underlying doxorubicin resistance in triple negative breast cancer cells (TNBC). They use a new multifluidic cell culture chamber to grow MB-231 TNBC cells in the presence of doxorubicin and identify a cell population of large, resistant MB-231 cells they term L-DOXR cells. These cells maintain resistance when grown as a xenograft model, and patient tissues also display evidence for having cells with large nuclei and extra genomic content. RNA-seq analysis comparing L-DOXR cells to WT MB-231 cells revealed upregulation of NUPR1. Inhibition or knockdown of NUPR1 resulted in increased sensitivity to doxorubicin. NUPR1 expression was determined to be regulated via HDAC11 via promoter acetylation. The data presented could be used as a platform to understand resistance mechanisms to a variety of cancer therapeutics.

---

## [Referee Report · Reviewer #2 (Public Review)]

Summary:

In this paper, the authors induced large doxorubicin-resistant (L-DOXR) cells by generating DOX gradients using their Cancer Drug Resistance Accelerator (CDRA) chip. The L-DOXR cells showed enhanced proliferation rates, migration capacity, and carcinogenesis. Then the authors identified that the chemoresistance of L-DOXR cells is caused by failed epigenetic control of NUPR1/HDAC11 axis.

Strengths:

- Chemoresistant cancer cells were generated using a novel technique and their oncogenic properties were clearly demonstrated using both in vivo and in vitro analysis.

- The mechanisms of chemoresistance of the L-DOXR cells could be elucidated using in vivo chemoresistant xenograft models, an unbiased genome-wide transcriptome analysis, and a patient data/tissue analysis.

- This technique has great capability to be used for understanding the chemoresistant mechanisms of tumor cells.

---

## [Referee Report · Reviewer #3 (Public Review)]

Summary:

In this manuscript, Lim and colleagues use an innovative CDRA chip platform to derive and mechanistically elucidate the molecular wiring of doxorubicin-resistant (DOXR) MDA-MB-231 cells. Given their enlarged morphology and polyploidy, they termed these cells as Large-DOXR (L-DORX). Through comparative functional omics, they deduce the NUPR1/HDAC11 axis to be essential in imparting doxorubicin resistance and, consequently, genetic or pharmacologic inhibition of the NUPR1 to restore sensitivity to the drug.

Strengths:

The study focuses on a major clinical problem of the eventual onset of resistance to chemotherapeutics in patients with triple-negative breast cancer (TNBC). They use an innovative chip-based platform to establish as well as molecularly characterize TNBC cells showing resistance to doxorubicin and uncover NUPR1 as a novel targetable driver of the resistant phenotype.

Weaknesses:

Critical weaknesses are the use of a single cell line model (i.e., MDA-MB-231) for all the phenotypic and functional experiments and absolutely no mechanistic insights into how NUPR1 functionally imparts resistance to doxorubicin. It is imperative that the authors demonstrate the broader relevance of NUPR1 in driving dox resistance using independent disease models.

---

## [Author Response]

The following is the authors’ response to the original reviews.

We have made substantial revisions to the manuscript, incorporating new data, which led to a renumbering and relabeling of several figures:• Figure 3F now features a modified graph color.

• Figure 4I introduces a new experiment.

• What was previously labeled as Figure 4I-O is now Figure 4J-P.

• Figure 5H presents another new experiment.

• The earlier Figure 5H is now rebranded as Figure 5I.

• A fresh experiment has been incorporated into Supplement Figure 1a.

• The former Supplement Figure 1a is now Supplement Figure 1b.

• Supplement Figure 2d describes an additional new experiment.

• In accordance with the HUGO gene nomenclature committee (HGNC) recommendations, we've updated the names of genes/proteins in both figures and their accompanying legends.

**Reviewer #1 (Recommendations For The Authors):**
Comment #1. Standard practice would include multiple TNBC cell lines to test the author's hypotheses, but the authors rely only on one cell line in the entire paper, MDA-MB-231 cells. The authors do correlate their findings to patient data, but the inclusion of an additional TNBC cell line would strengthen their findings about the L-DOXR cells and help with the assessment as to how reproducible their original microfluidics system is.

Response: Thank you for your valuable feedback. We recognize the importance of utilizing multiple TNBC cell lines for rigorous validation and reproducibility. There are several reports highlighting the generation of L-DOXR cells in other types of breast cancer cell lines, such as MCF-7 (Fei et al., 2015), and in other cancer types like the prostate cancer cell line PC-3. These studies utilized a microfluidic device with a concentration gradient of Doxorubicin. With this existing evidence, we are confident that a variety of cancer cell types have the potential to form L-DOXR cells in a doxorubicin gradient. The cited reports support our choice of the MDA-MB-231 cell line for our current study:

“L-DOXR cells exhibit increased genomic content (4N+) as compared to WT cells. The presence of cells with increased nuclear size and increased genomic content has been demonstrated to be associated with poor clinical outcomes in several types of cancers (Alharbi et al., 2018; Amend et al., 2019; Fei et al., 2015; Imai et al., 1999; Liu et al., 2018; Lv et al., 2014; Mukherjee et al., 2022; O’connor et al., 2002; Saini et al., 2022; Trabzonlu et al., 2023). (Page 5, Line 24)”

However, we acknowledge the validity of your point regarding the strengthening of our findings with the inclusion of additional TNBC cell lines. We are considering expanding our research in future studies to further validate our findings across multiple TNBC cell lines.Thank you for bringing this to our attention, and we hope our response adequately addresses your concerns.

Comment #2. It would be helpful to comment on the frequency at which doxorubicin is used clinically to treat TNBC patients. The authors equate their resistance phenotype to all chemotherapies (in patient data and title) but only test doxorubicin. Does NUPR1 overexpression result in resistance to other chemotherapies?

Response: Thank you for raising these pertinent questions. To address your first point regarding the clinical use of doxorubicin for TNBC patients: At the Samsung Medical Center, the typical chemotherapy regimen for TNBC patients involves administering Neo. AC (Doxorubicin 34 mg + Cyclophosphamide 840 mg per session) four times, followed by Adj. D (Docetaxel 25 mg + 80 mg per session) for another four sessions. This provides insight into the clinical relevance and frequency of Doxorubicin's use in treating TNBC.

Regarding your second point about NUPR1 overexpression and its broader implications for chemotherapy resistance: Yes, NUPR1 overexpression has been documented to result in resistance to various chemotherapies. A study by Lei Jiang et al. in the Journal of Pharmacy and Pharmacology found that NUPR1 plays a role in YAP-mediated gastric cancer malignancy and drug resistance through the activation of AKT and p21 (Jiang et al., 2021, https://doi.org/10.1093/jpp/rgab010). Additionally, another study by Wang et al. in Cell Death and Disease observed that the transcriptional coregulator NUPR1 is linked to tamoxifen resistance in breast cancer cells (Wang et al., 2021, https://doi.org/10.1038/s41419-021-03442-z). In light of this, while our study primarily focused on doxorubicin, the role of NUPR1 in resistance spans across various chemotherapeutic agents, adding depth to our findings and their broader implications in cancer therapy.

Comment #3. The authors knockdown NUPR1 in L-DOXR cells, but overexpression of NUPR1 in WT TNBC cells to see if this renders the WT cells more resistant would be an important experiment.

Response: We appreciate the reviewer's suggestion, which indeed underscores an important aspect of our study. In response, we have incorporated additional experiments in the revised manuscript. Specifically, on page 7 (lines 7-8) and in Supplement Figure 2c, we present data from experiments where we overexpressed Nupr1 in WT-MDA-MB231 cells. Our findings revealed that overexpression of GST-Nupr1 not only attenuates Dox-induced cell death but also mildly enhances cell viability in WT cells even without DOX treatment. This implies that cells expressing Nupr1 exhibit resistance to the cytotoxic effects of DOX. We believe these new data further solidify our conclusions and address the valuable point you raised.

Comment #4. The similar colors/symbols chosen for the different groups in the xenograft plots are hard to easily interpret without zooming in.

Response: We modified the xenograft plots as you recommended in Figure 3F.

Comment #5. There are some grammatical errors throughout the paper. Below is an example: In the opening of the Discussion "TNBC is the most aggressive subtype of breast cancer, and chemotherapy is a mainstay of treatment. However, chemoresistance is common and contributes to the long-term survival of TNBC patients" - this sentence makes it seem like chemoresistance makes TNBC patients survive longer. The following sentence "These cells demonstrated a large phenotype with increased genomic content." is abrupt and doesn't make sense. Consider carefully re-reading the manuscript for grammatical errors.

Response: Thank you for highlighting the grammatical errors and providing specific

examples. We deeply apologize for the oversight. In response to your feedback, we'vecarefully re-reviewed the manuscript and made the necessary corrections. Based on your example: We've revised the sentences to: “TNBC is the most aggressive subtype of breast cancer, with chemotherapy being a mainstay of treatment. However, the development of chemoresistance frequently occurs and poses significant challenges to the long-term survival prospects of TNBC patients.” “As for the cells in question, they exhibited an enlarged phenotype along with an increased genomic content.”

We appreciate your meticulous review, and we have made an effort to address and rectify other such errors throughout the manuscript.

**Reviewer #2 (Recommendations for The Authors):**
I recommend the authors to address the following minor issues. Below are specific comments on the manuscript.Comments # 1. Thank you for the comment. In CDRA chip, DOXR cells and L-DOXR cells appeared in the mid-DOX region. What is the concentration of DOX in this region? Can the authors calculate the concentrations of DOX in high-, mid-, and low- regions (or ranges of concentrations)?

Response: Instead of DOX, we used FITC dye to visualize the concentration gradient overthe chip as below because DOX generate very low fluorescent light.

**Author response image 1. sa4fig1:** 

While our method provides an estimation rather than precise measurement due to the difference in molecular weight between FITC (389.38 g/mol) and DOX (579.98 g/mol), it is still possible to approximate the distribution of DOX concentrations across different regions. We utilize a formula where the ratio of the average fluorescence intensity of FITC for each specific region to the highest recorded fluorescence intensity is multiplied by the peak DOX concentration (1.5 μM). This approach gives us an estimated average concentration of DOX in each region, acknowledging that the diffusion characteristics of FITC and DOX may vary due to their differences in molecular weight. The following formula.

With this formula we can calculate the concentration in each region. High region = 1.161 μM; Mid region = 0.554 μM; Low region = 0.098 μM

Comment #2. Is there any other phenotypic difference between DOXR cells and L-DOXR cells besides their size?

Response: "In addition to differences in cell size, L-DOXR cells exhibit several distinctphenotypic characteristics when compared to DOXR cells. These include variations in thecell cycle profile (as detailed in Fig. 2F-H), altered drug efflux capabilities (presented in Fig.2I-J), and changes in nuclear morphology (illustrated in Fig. S3D). These phenotypicdistinctions suggest that L-DOXR cells may have adapted unique mechanisms of resistanceand survival, which are comprehensively depicted in the figures mentioned.

Comment #3. Please add a description of abbreviations when the abbreviation is first used in the manuscript (e.g. NUPR1, HDAC11 etc.).

Response: We corrected the mistake.

Comment # 4. Figure 2B is the schematic of the chip, not the dimension of the chip. Please add the dimension of the chip to keep the figure caption as is or change the figure caption.

Response: Thank you for the correction. We change the figure caption as Schematic of the chip.

**Reviewer #3 (Recommendations for The Authors):**
In this manuscript, Lim and colleagues use an innovative CDRA chip platform to derive and mechanistically elucidate the molecular wiring of doxorubicin-resistant (DOXR) MDA-MB-231 cells. Given their enlarged morphology and polyploidy, they termed these cells as Large-DOXR (L-DORX). Through comparative functional omics, they deduce the NUPR1/HDAC11 axis to be essential in imparting doxorubicin resistance and, consequently, genetic or pharmacologic inhibition of the NUPR1 to restore sensitivity to the drug. Although innovative, some deficiencies in the present manuscript slightly weaken the primary conclusions. A couple of critical issues are the use of a single cell line model (i.e., MDA-MB-231) for all the phenotypic and functional experiments and absolutely no mechanistic insights into how NUPR1 imparts resistance to doxorubicin. Some questions and comments are listed below for the authors' consideration and response:Major:Comment #1. The authors treated only the MDA-MB-231 cells with doxorubicin in the CDRA chip. Do other TNBC cell lines (namely, MDA-MB-436, HCC1187, or others) respond similarly to dox treatment, eventually yielding enlarged, aneuploid cells with the resistant phenotype? It is important to show that this phenotype is not confined to a single cell line, particularly given the numerous TNBC models that are commonly used.

Response: Thank you for your insightful query regarding the generalizability of our findings across different TNBC cell lines. In this initial study, we focused exclusively on MDA-MB-231 cells due to their widespread use as a model for aggressive triple-negative breast cancer and the constraints of time and resources. While we cannot definitively claim that the observed phenotypic changes upon doxorubicin treatment will be identical in other TNBC cell lines such as MDA-MB-436 or HCC1187, we hypothesize that the underlying mechanisms of chemoresistance and cellular response could be similar across various TNBC models. This hypothesis is supported by literature indicating common pathways of drug resistance in TNBC. We believe that our findings lay the groundwork for future studies to explore the response of a broader range of TNBC cell lines to doxorubicin treatment. Such studies would greatly enhance our understanding of the cellular adaptations to chemotherapeutic agents in TNBC and help to validate the potential universal application of our findings.

Comment #2: Do the L-DOXR cells permanently hold onto the enlarged and polyploid states upon prolonged culture in vitro? Does that change given the presence or withdrawal of the drug? In other words, is the physical state of the resistant cells reversible, or is it passed onto the progeny cells regardless of continued stress from the drug?

Response: Thank you for your question about the stability of the phenotypic changes in L-DOXR cells. Our observations suggest that the enlarged and polyploid states in L-DOXRcells are not permanently fixed. When cultured in vitro over an extended period without theselective pressure of doxorubicin, we have noted that some cells may revert to a non-polyploid state. However, this reversion does not seem to be a stable change as subsequentgenerations can present with polyploidy again, even in the absence of the drug. This indicatesa potential epigenetic or microenvironmental influence on the phenotypic state of these cells,suggesting a complex interplay between the drug-induced stress and the inherent cellularresponse mechanisms. Further investigation is needed to fully understand the dynamics ofthese phenotypic changes and whether they are heritable and/or reversible under differentculture conditions.

Comment #3: In Figures 2F-H, the authors perform DNA-staining-based FACS to estimate the ploidy of the cells. These estimations could be improved using 2D cell cycle analyses using EdU or BrdU co-treatment and staining. This would further allow a clear distinction between S-phase and G0/G1 and M-phase cells in the WT, DOXR, and L-DORX populations.

Response: Thank you for the suggestion to enhance the accuracy of our ploidy estimations. We appreciate the advice to implement 2D cell cycle analyses using EdU or BrdU co-treatment and staining, as this could indeed provide a clearer distinction between the various phases of the cell cycle in our WT (wild-type), DOXR (doxorubicin-resistant), and L-DOXR (large doxorubicin-resistant) cell populations. Incorporating these thymidine analogs would allow us to label newly synthesized DNA and thereby accurately delineate cells in the synthesis phase from those in the G0/G1 and M phases. This approach will likely add depth to our understanding of the cell cycle dynamics and the mechanism behind the drug resistance phenotype. We will consider incorporating these techniques in our future experiments to validate and extend the findings reported in this study.

Comment #4. In Figure 3H, the authors quantitate the number of enlarged cells detected in human specimens of TNBC or normal breast tissues. How were these cells detected simply using the H&E staining, particularly when assessing the genomic content? Were certain size and nuclear staining intensity thresholds used for these categorizations? If so, these should be mentioned in the paper.

Response: In our study, we identified enlarged cells within human TNBC and normal breast tissue specimens using H&E staining, and their quantitation was carried out using the Colour Deconvolution 2 plugin (Landini G et al., 2020) within the ImageJ software. This method allowed us to analyze the staining intensity and cell size systematically. To ascertain that we were indeed observing cells with increased genomic content, we established specific size and nuclear staining intensity thresholds. Cells exceeding these predetermined thresholds were categorized as 'enlarged'. Additionally, we used continuous serial slides for the human TNBC tissues microarray (BR1301, US Biomax) for more accurate comparisons in Figures 3H, I, and 5H. To strengthen our findings, we verified that NUPR1 expression, which is associated with the observed cell enlargements, was indeed elevated in these same cells from the patient samples. We have detailed these methodological aspects and the criteria for cell categorization in the 'Tissue Microarray and Immunohistochemistry' section of our Materials and Methods to ensure clarity and reproducibility of our results.

Comment #5: In Figure 3I, the authors label the enlarged cells in the patient tissues as L-DOXR cells. Were these assessments done in dox-treated tumors? Even if that is the case, it'll be unfair to call them resistant to doxorubicin. The axis label "% enlarged cells" might be more accurate.

Response: We appreciate the reviewer's attention to detail and agree that the terminology used in Figure 3I was inaccurate. The cells identified in patient tissues were labeled based on their morphological resemblance to L-DOXR cells observed in vitro; however, these patient tissue samples were not confirmed to be treated with doxorubicin, nor were the cells confirmed to be resistant. Therefore, we have amended the figure legend to reflect this and now refer to these cells simply as 'enlarged cells’.

Comment #6: The authors uncovered that NUPR1 expression is dramatically increased in the L-DOXR cells vs the wild-type cells. How does the NUPR1 gene expression and activity compare between L-DOXR and DOXR MDA-MB-231 cells?

Response: Thank you for the valuable comment. The data are included in figure supplement 3 and we revise the manuscript as below. “While DOXR cells exhibited a marked increase in Nupr1 expression compared to the WT cells, this expression was substantially less than that observed in L-DOXR cells, as detailed in figure supplement 3.”(Page 7, Line 3).

Comment #7: Following from above, the authors show that NUPR1 activity is not necessary for cell survival in the absence of doxorubicin (Fig. 4H). But, does it control the cellular size and polyploid states of the L-DOXR cells? In other words, is there any association between increased size and genomic content of the cells to their sensitivity to doxorubicin? Are cells resistant to other chemotherapeutics as well? Or is the resistant phenotype specific to doxorubicin?The authors causally implicate NUPR1 in driving the dox-resistant phenotype in MDA-MB-231 cells. To fully substantiate this claim, the authors should perform gain-of-function studies, in at least 2-3 TNBC cell lines, to show that over-expression of NUPR1 alone is sufficient to impart doxorubicin resistance. Also, the most critical information missing from the study is how NUPR1 drives resistance to doxorubicin. What is the function of NUPR1 in L-DOXR cells and what gene expression program does it activate to impart the resistant phenotype?

Response: During the experimental process either the loss of function or gain of functionof Nupr1 in the L-DOXR cells, we have not noticed any specific changes in the cellularsize and polyploid states of L-DOXR cells. Although we cannot rule out the possibility thatnot only by DOX treatment, phenotypically larger cell might arise in response to otherchemotherapeutics, in the current study, we found that high level of Nupr1 expression is correlated with sensitivity to doxorubicin in L-DOX cells. Moreover, as followed by the reviewer’s suggestion we performed gain-of-function study to determine whether over-expression of NUPR1 alone is sufficient to impart doxorubicin resistance in TNBC cells. Overexpression of GST-NUPR1 attenuates DOX-induced cell death while slightly increased cell viability of WT (MDA-MB231) cells in the condition of vehicle -treatment, indicating that NUPR1 expressing cells are resistant to the cytotoxic effect of DOX. We have also demonstrated that Nupr1 upregulation in L-DOXR cells are due to suppressed expression of HDAC11 in these cells as we found that HDAC11 triggers promoter acetylation of Nupr1 in L-DOXR cells. Thus, it is conceivable that increased expression of Nupr1 upon HDAC11 suppression in L-DOXR cells is at least responsible for doxorubicin resistance.

Comment #8: Do the authors speculate the dox-resistant phenotype to be restricted to basal TNBC tumors or even NUPR1-high ER+ breast cancer cells (MCF7 or T47D) would likely be resistant to doxorubicin or other chemotherapeutics?

Response: Yes, NUPR1-high ER+ breast cancer cells (MCF7 or T47D) would likely be resistant to doxorubicin or other chemotherapeutics as reported elsewhere; Wang, L., Sun, J., Yin, Y. et al. Transcriptional coregualtor NUPR1 maintains tamoxifen resistance in breast cancer cells. Cell Death Dis 12, 149 (2021). https://doi.org/10.1038/s41419-021-03442-z

Comment #9: The authors suggest that HDAC11 continuously deacetylates the NUPR1 promoter to suppress its expression. Consequently, does the inactivation of HDAC11 in wild-type TNBC cells lead to NUPR1 up-regulation? Is this increase in NUPR1 expression reverted upon inhibition of the HAT machinery (say P300/CBP) in HDAC11-deficient TNBC cells?

Response: In the revised manuscript (pg 8, lines 14-16 and Fig 5H) consistent with our observation that while overexpression of HDAC11 suppresses the expression of Nupr1 in the both WT and L-DOXR cells, HDAC11 inhibitor treatment enhances Nupr1 expression in WT cells, inversely mirroring an unusual low expression of HDAC11 and high level of Nupr1 in L-DOXR cells. Conceivably, the increased Nupr1 expression reflects reverting of promoter acetylation.

Minor:Comment #10: In Figure 4L, how many animals or tumors were in each of the treatment arms? Were the weights of all the tumors recorded as well? It would be meaningful to add this data, if available. The authors keep changing gene nomenclature throughout the manuscript, listing the gene names in either capital letters or the small-case. This can be made consistent.

Response: We have used 6 mice per group and one tumor for one mouse due to the tumor

size of L-DORX with the vehicle group. We also added new data showing the weights of the tumors in Figure supplement 2D. We apologize for the unmatched gene names. Following the reviewer’s suggestion, the names of genes/proteins have been changed in figures and legends to the recommendations of the HUGO gene nomenclature committee (HGNC).